# Optimization and Application of Real-Time qPCR Assays in Detection and Identification of Chlamydiales in Products of Domestic Ruminant Abortion

**DOI:** 10.3390/pathogens12020290

**Published:** 2023-02-09

**Authors:** Annelize Jonker, Anita L. Michel

**Affiliations:** Tuberculosis and Brucellosis Research Programme, Department of Veterinary Tropical Diseases, Faculty of Veterinary Science, University of Pretoria, Pretoria 0002, South Africa

**Keywords:** 1, Chlamydiales 2, abortion 3, ruminant 4, *Chlamydia 5*, *Parachlamydia 6*, *Waddlia*

## Abstract

Domestic ruminant abortions due to infectious agents represent an important cause of economic losses in the agricultural industry. This study aimed to optimise and apply qPCR assays for detection of Chlamydiales in domestic ruminant abortion cases. Primers and probes for detection of the order Chlamydiales, *Chlamydia abortus*, *Chlamydia pecorum*, *Parachlamydia acanthamoeba* and *Waddlia chondrophila* were taken from the literature to create one singleplex and two duplex assays and the assays were optimised. Placentitis and pneumonia are pathological lesions associated with Chlamydiales infection. In a previous study, twenty-five clinical cases had pathological lesions of placentitis or pneumonia. These cases were investigated further by application of the qPCR assays in this study. Chlamydiales were detected in 16 cases. *C. abortus*, *P. acanthamoeba* and *W. chondrophila* were detected in bovine; and *C. pecorum* and *W. chondrophila* in ovine and caprine cases. Chlamydiales were detected in three previously inconclusive cases. Identification was improved from genus to species level (*C*. *pecorum*). Four cases remained inconclusive. In conclusion, detection of Chlamydiales and differentiation to species level was improved. This study reports the first detection of *P. acanthamoeba* and *W. chondrophila* in abortion cases in South Africa, indicating a potentially significant role in abortions in this country.

## 1. Introduction

Infectious agents are the most common causes of domestic ruminant abortions [1] representing important economic losses in the agricultural industry [2]. Abortion is defined as the premature end of a pregnancy with the expulsion of a non-viable foetus [3]. Three families in the order Chlamydiales are of significance as agents of abortion, namely Chlamydiaceae, Waddliaceae and Parachlamydiaceae [4]. Chlamydiales are adapted to a very broad spectrum of hosts and are agents of a variety of diseases in animals and humans [4]. Several members of the order are of zoonotic significance. *Chlamydia abortus* (*C. abortus*), *Waddlia chondrophila* (*W. chondrophila*) and *Parachlamydia acanthamoebae* (*P*. *acanthamoebae*) have been detected in human cases of miscarriage and pneumonia [5,6,7]. 

Members of the order Chlamydiales are small, gram negative, obligate intracellular bacteria with a biphasic developmental cycle [4]. Faeces, urine, placenta and/or discharges from the nose, eyes or vulva of an infected animal may serve as sources of infection [8,9,10]. A primary infection is established in the tonsils following which the bacteria may disseminate via blood or lymph to the placenta [4]. 

The most important family is the Chlamydiaceae to which *Chlamydia* species belong [11]. *Chlamydia abortus*, the cause of ovine enzootic abortion (OEA), can also infect cattle and goats and is among the most common bacterial causes of ruminant abortion worldwide [4]. It can become endemic in herds [11] and results in late-term abortion or weak full-term calves, lambs or kids [10,12,13]. Studies suggest that *Chlamydia pecorum* (*C. pecorum*) is endemic in cattle, sheep and goats worldwide [9,14]. It is often carried in ruminant gastrointestinal tracts, but only causes sporadic abortions [9,14]. An Australian study by Clune et al. [15] indicated that it could be an important cause of abortion in that country. Development of detection methods and research into the significance of *C. pecorum* in domestic ruminants has increased in recent years in other countries [9]. However, little is reported on these bacteria in South Africa. 

Other Chlamydiales of interest are *Waddlia chondrophila* (family Waddliaceae) [11] and *Parachlamydia acanthamoebae* (family Parachlamydiaceae). Both species were implicated in bovine abortions in several studies in the Northern Hemisphere [10,16,17,18]. *P. acanthamoeba* was detected in material form ovine abortion cases in a Tunisian study by Barkallah et al. [19]. No literature on *P. acanthamoebae* or *W. chondrophila* detection in the Southern Hemisphere was found. The reason may be that routine abortion investigations do not commonly include detection of these bacteria [4,10]. The role that *P. acanthamoebae* and *W. chondrophila* play in abortion worldwide needs to be investigated further [10,20]. 

Chlamydiales infections often lead to placentitis and bronchopneumonia in the foetus [10,21]. *Chlamydia abortus*, *C. pecorum* and *W. chondrophila* infections lead to a purulent and/or necrotising placentitis, often including vasculitis [4,13,16,21,22]. *P. acanthamoeba* infection leads to a purulent and/or necrotising placentitis, most commonly without vasculitis [22,23]. Both *Parachlamydia* species and *W. chondrophila* infection have been associated with pneumonia in the foetus [16,24].

Members of the Chlamydiales fall in the difficult-to-culture category [25]. Historically, isolation in cell culture or chicken embryos is the ‘gold standard’ [26]. Due to the low sensitivity, conventional cell-culture methods are seldom used and molecular-based methods are investigated for detection of these bacteria [27]. Cases of bovine, ovine and caprine abortion where the aetiology remains unknown, but the presence of an infectious agent is indicated by a purulent or necrotising placentitis, or septicaemia, are a problem encountered worldwide [2,20]. Foetuses that died 24 to 48 h before abortion may already be autolytic by the time they are aborted [12]. This presents an additional challenge as autolysis masks lesions and the agent of abortion may be outcompeted by other organisms [28]. These challenges have prompted the application of methods that are less affected by autolysis. 

Immunohistochemical staining methods (IHC) use antibodies directed against chlamydial lipopolysaccharides or other surface antigens to demonstrate these bacteria in cells. However, these methods are often not specific for a bacterial species [10,14]. 

Molecular analysis offers more sensitive and specific methods in the form of conventional or quantitative real-time PCR (qPCR). These methods offer the opportunity to improve the diagnostic rate of traditional methods, such as smear examination and culture [29]. However, sensitivity and specificity differ between different PCR protocols. Real-time PCR methods provide increased sensitivity compared with conventional PCR, and qPCR methods provide quantification of target DNA [30]. A study in Switzerland reported an improvement in the overall diagnostic rate of 51.9% when qPCR was employed in abortion investigation in addition to conventional culture [31]. Duplexing is the simultaneous amplification and quantification of two target sequences in a single qPCR assay [20]. This approach increases sample throughput, reduces labour, minimises the impact of pipetting errors, and saves on reagent costs [20]. 

A screening PCR for the order Chlamydiales followed by species-specific PCRs is valuable for detection of *Chlamydia* and *Chlamydia*-like bacteria in products of abortion and swabs of the genital tract [17,32]. A Chlamydiales-specific real-time qPCR (pan-Chlamydiales qPCR) with locked nucleic acids targeting the conserved 16S rRNA gene was published by Lienard et al. [32]. This PCR was successfully applied to screen human and cattle samples for Chlamydiales [17,32,33]. In the study by Wheelhouse et al. [33], amplicons from PCR-positive cattle abortion samples were sequenced and revealed *C. abortus*, *P. acanthamoebae* and *Neochlamydiae hartmannellae*. In the Tunisian study by Barkallah et al. [17], further analysis of amplicons from PCR-positive cattle vaginal swab samples by PCR together with sequencing led to the identification of *C. abortus*, *C. pecorum*, *P. acanthamoeba* and *W. chondrophila*. As vaginal swab samples were used, the bacteria detected could not be correlated with placental or foetal lesions to determine their role in abortions. 

Species-specific singleplex qPCR assays were developed to detect the ompA gene of *C. abortus* [34] and *C. pecorum* [35], a fragment of the 16S rRNA gene of *W. chondrophila* [27] and of *P. acanthamoeba* [36]. These qPCR assays were successfully used to detect DNA of *C. abortus*, *C. pecorum, P. acanthamoeba* and *W. chondrophila* in animal samples [17,22,34,35,37]. 

Different multiplex PCR assays for detection of Chlamydiales and other bacteria in products of abortion are reported in the literature [20,38,39,40]. In addition, multiple potential agents of abortion are often reported [17,20,31]. This raises the question of the role mixed infections play in abortion. In order to differentiate between infection and colonisation by *C. abortus*, a study by Gutierrez et al. [37] established a diagnostic cut-off point (DNA copies: ≥10^6^) for the duplex qPCR they used.

The aim of this study was to optimise and apply singleplex screening and duplex species-specific qPCR assays for detection of Chlamydiales DNA in products of domestic ruminant abortions, where placentitis and/or pneumonia was previously reported as pathological lesion. The use of a diagnostic cut-off point to determine the clinical significance of *C. abortus* results was investigated.

## 2. Materials and Methods

In this descriptive study, synthetic controls as well as DNA from bacterial cultures and ruminant abortion cases were used to optimise three qPCR assays for the detection of Chlamydiales DNA in products of ruminant abortion cases. The qPCR assays were optimised by means of a two-phase test validation process. During the first phase, analytical test validation comprised the following steps: (a) testing synthetic-DNA-positive controls of the organisms of interest (sensitivity), (b) testing pure cultures of non-target organisms that are commonly found in products of abortion (specificity), (c) testing extracted DNA of foetal stomach content, lung and placenta spiked with synthetic DNA controls of the pathogens of interest to determine limit of detection. During the second phase, diagnostic test validation comprised: (a) parallel testing of field abortion samples where the causative pathogen was detected previously by histopathology, immunohistochemistry or PCR to demonstrate diagnostic sensitivity and (b) testing of foetal tissue/placenta of known uninfected animals to demonstrate diagnostic specificity. 

### 2.1. Synthetic Controls

Synthetic oligonucleotides were used for determination of specificity, sensitivity and limit of detection of the qPCR assays. Synthetic oligonucleotides (gBlocks^®^ Gene Fragments) were obtained from Integrated DNA Technologies, Coralville, IA, USA (www.idtdna.com, accessed on 20 November 2017) (IDT). The synthetic oligonucleotides were designed to contain three regions, the forward and reverse primers, and the probe of each target. The design was based on genomic reference sequences obtained from Genbank (http://www.ncbi.nlm.nih.gov/GenBank/index.html, accessed on 6 February 2022): Nr 036834.1 *Chlamydia abortus* strain Ov/B577 AY601755.1 *Chlamydophila abortus* clone 5 OmpA gene, KX388207.1 *Chlamydia pecorum* isolate M17/Ocular, Nr 074886.1 *Waddlia chondrophila* WSU86.1044 and Nr 026357.1 *Parachlamydia acanthamoeba* Strain B179. The synthetic controls were reconstituted according to the manufacturer’s instructions.

### 2.2. Microorganisms

Specificity of the PCR assays was tested using DNA extracted from different non-target bacteria often associated with products of abortion. Live intracellular control strains *C. abortus* (ATCC VR-165), *P. acanthamoeba* (ATCC VR-1476) and *W. chondrophila* (ATCC VR-1470) were procured from the American Type Culture Collection, Manassas, VA, USA. These bacteria were cultivated in five millilitres serum-casein-glucose-yeast extract medium (SCGYEM) at 30 °C for 6 days with *Acanthamoeba castellani* (ATCC 50739) as host according to ATCC instructions. Cultures were then harvested and centrifuged at 5000× *g* for 5 min. The supernatant was discarded and the pellet was resuspended in one millilitre of sterile PCR grade water. 

Live bacterial control strains that grow on acellular media (Table 1) were streaked on 5% blood agar plates and incubated in 5% CO_2_ at 37 °C until growth was observed. Bacterial suspensions were made in 5% saline to a turbidity equal to a 0.5 McFarland standard. Fifty microlitres of *Acholeplasma laidlawlii* (NCTC 10116) suspension was inoculated in *Mycoplasma* broth (Oxoid CM0403 & SR0059, Thermoscientific, Gauteng, South Africa) and incubated in normal air at 37 °C until the broth was turbid. The bacterial suspensions and *Acholeplasma laidlawlii* broth culture were then diluted 1:100 in sterile PCR grade water. 

### 2.3. Clinical Samples

All 25 clinical bovine, ovine and caprine abortion cases where necrotic placentitis and/or pneumonia were reported were selected from the 135 clinical abortion cases in the same study [41]. Fifty samples (17 placentas, 19 stomach contents, 14 lungs) from these 25 cases were analysed in this study. 

DNA from 49 samples (14 placenta, 17 stomach content, 18 lung) from clinical ruminant abortion cases (13 bovine, 9 ovine, 10 caprine), which were available from a previous study [41], were used in specificity and limit of detection analyses. Only samples that had pathological lesions other than placentitis or pneumonia, or no lesions were selected.

### 2.4. DNA Extraction

DNA was extracted according to the manufacturer’s instructions from all culture suspensions and the *Acholeplasma* broth culture (500 μL) and clinical samples (25 mg) using the NucleoSpin^®^ Tissue kit (Macherey-Nagel GMBH & Co., Düren, Germany). 

### 2.5. Primers and Probes for Detection of Chlamydiales

Primers and probes were selected from published studies according to base pair size and annealing temperature [27,32,34,35,36]. The probes for Pan-Chlamydiales, *W. chondrophila* and *P*. *acanthamoeba* were slightly modified by replacing locked nucleic acids by a Minor Groove Binder (MGB) probe. Primers were synthesised by Integrated DNA Technologies, Coralville, IA, USA (www.idtdna.com, accessed on 20 November 2017) and probes by Thermo Fisher Scientific, Waltham, MA, USA (www.thermofischer.com, accessed on 13 November 2017). Sequences and targets for primers and probes are as in Table 2.

**Table 2 pathogens-12-00290-t002:** Sequences, target genes, base pair sizes and concentration of primers and probes (For: Forward; Rev: Reverse; P: Probe; FAM: 6-carboxy-fluorescein; TAMRA: 6-carboxy-tetramethylrhodamin; VIC: 2′chloro-7phenyl-1,4-dichloro-6-carboxy-fluorescein; MGB: Minor Groove Binder).

Chlamydiales	Target Gene	Primer and Probe Sequences (5′ to 3′)	Base Pair Sizes	Concentration (μM)	References
		**Singleplex qPCR assay**			
Pan-Chlamydiales	16S rRNA	For CCGCCAACACTGGGACT	207 to 215	0.1	[32]
		Rev GGAGTTAGCCGGTGCTTCTTTAC		0.1	
		P VIC-CTACGGGAGGCTGCAGTCGAGAATC-MGB		0.1	
		**Multiplex qPCR assays**			
Assay 1 *Chlamydia abortus*	*omp*A	For GCAACTGACACTAAGTCGGCTACA	82	0.9	[34]
		Rev ACAAGCATGTTCAATCGATAAGAGA		0.9	
		P FAM-TAAATACCACGAATGGCAAGTTGGTTTAGCG-TAMRA		0.2	
*Chlamydia pecorum*	*omp*A	For CCATGTGATCCTTGCGCTACT	76	0.9	[35]
		Rev TGTCGAAAACATAATCTCCGTAAAAT		0.9	
		P VIC-TGCGACGCGATTAGCTTACGCGTAG-TAMRA		0.2	
Assay 2 *Waddlia chondrophila*	16SrRNA	For GGCCCTTGGGTCGTAAAGTTCT	101	0.5 (This study)	[27]
		Rev CGGAGTTAGCCGGTGCTTCT		0.5	
		P VIC-CATGGGAACAAGAGAAGGATG-MGB		0.2	
*Parachlamydia acanthamoebae*	16SrRNA	For CTCAACTCCAGAACAGCATTT	103	0.5 (This study)	[36]
		Rev CTCAGCGTCAGGAATAAGC		0.5	
		P FAM-TTCCACATGTAGCGGTGAAATGCGTAGATATG-MGB		0.2	

The Pan-Chlamydiales qPCR was used as a singleplex assay. The *C. abortus* and *C. pecorum* qPCRs were combined in a duplex qPCR assay, while the *P. acanthamoeba* and *W. chondrophila* qPCRs were combined in a second duplex qPCR assay. 

### 2.6. Quantitative Real-Time PCR (qPCR)

Amplification and detection of PCR products were performed in clear PCR strip tubes with caps (Applied Biosystems, Carlsbad, CA, USA) using a StepOnePlus^®^ Real Time PCR system (Applied Biosystems, Carlsbad, CA, USA). The final volume of reaction mixtures was 20 μL. Pan-Chlamydiales qPCR primer and probe concentrations were as published by Lienard et al. [32]. The *C. abortus* and *C. pecorum* qPCR primer and probe concentrations were used as published [34,35]. In this study, *P. acanthamoeba* and *W. chondrophila* qPCR primer and probe concentrations were increased to 0.5 and 0.2 µM, respectively. The reaction mixture was completed by addition of 4 μL 5x Hot FirePol Probe Universal qPCR mix (Solis BioDyne, Tartu, Estonia), 10 μL ultrapure water and 2.5 μL DNA sample. Cycling conditions were: 95 °C for 10 min (initial activation), and 40 cycles of 95 °C for 15 s (denaturation) and 60 °C for 1 min (annealing/elongation).

### 2.7. Analytical Sensitivity 

PCR amplification efficiency was established by standard curves where the amplification efficiency is calculated from the slope of the log-linear portion of the standard curve [42,43]. Reconstituted synthetic controls were diluted to 10^8^ DNA copies/μL. Then 1:10 serial dilutions were prepared to 10 DNA copies/μL. Analytical sensitivity was determined in triplicate. 

### 2.8. Analytical Specificity

Analytical specificity for each qPCR assay was evaluated using three Chlamydiales and nine non-Chlamydiales bacteria that are likely to occur as contaminants in products of abortion. The *Acanthamoeba castellani* ATCC control was included since it was used as a host for cultivating the live Chlamydiales controls.

### 2.9. Limit of Detection of the qPCR Assays

Samples of stomach content, placenta and lung from cases where no placentitis or pneumonia was reported were analysed with the qPCRs in this study. DNA from samples where there was no amplification of target DNA was spiked with 10, 5, 2 and 1 DNA copies (*C. abortus*/*C. pecorum* & *P. acanthamoeba*/*W. chondrophila*) or 1000, 500, 250 and 62 DNA copies (Pan-Chlamydiales) of the synthetic controls (Integrated DNA Technologies, Coralville, IA, USA). Six repetitions of the spiked samples were analysed to determine detection limits of the singleplex and duplex qPCR assays. 

### 2.10. Diagnostic Sensitivity

Diagnostic sensitivity is the ability of a test to detect samples identified by a reference method as positive. It is calculated as follows: TP/(TP + FN), where TP = true positive and FN = false negative [43]. Samples of placenta, stomach content and lung from cases where placentitis and/or pneumonia was observed previously by histopathology were analysed by the qPCR assays in this study and the results were used to calculate diagnostic sensitivity. 

### 2.11. Diagnostic Specificity

Diagnostic specificity is the ability of a test to identify samples that are found to be negative by a reference method. The formula is TN/(TN + FP), where TN = true negative and FP = false positive [43]. Samples of placenta, stomach content and lung from cases where no placentitis or pneumonia was observed previously by histopathology were analysed by the qPCR assays in this study. 

### 2.12. Quality Assurance

A β-actin internal amplification control (Taqman^®^ Gene Expression Assays, Applied Biosystems, Carlsbad, CA, USA) was included to assess the efficiency of the DNA extraction process and presence/absence of DNA polymerase inhibitors. Nuclease-free water was included as a negative control.

### 2.13. Data Analysis

Data analysis was performed with Applied Biosystems software (version 2.3) in the StepOnePlus^®^ Real Time PCR system as well as Microsoft Excel 2016. 

## 3. Results

### 3.1. Sensitivity, Specificity, Efficiency, and Limit of Detection of qPCR Assays for Detection of Chlamydiales

#### 3.1.1. Analytical Sensitivity

Standard curves for the Pan-Chlamydiales, *C. abortus*/*C. pecorum* and *P. acanthamoeba*/*W. chondrophila* are represented in Figure 1, Figure 2 and Figure 3. The data for the singleplex Pan-Chlamydiales qPCR assay were not linear, which had a negative influence on correlation coefficient (R^2^) and efficiency calculations. An R^2^ of 0.9585 and an efficiency of 104.432% were calculated. The R^2^ and efficiency were lower than the *C. abortus/C. pecorum* duplex assay, but higher than the *P. acanthamoeba/W. chondrophila* duplex assay. The R^2^ values in the *C. abortus*/*C. pecorum* duplex assay were slightly better than the R^2^ of the singleplex qPCRs. Efficiency in the duplex assay was 99.5% (*C. abortus*) and 99% (*C. pecorum*). Multiplexing of *P. acanthamoeba* and *W. chondrophila* qPCR assays resulted in data that were not linear, also with a negative influence on R^2^ and efficiency calculations. Efficiency of the *W. chondrophila* qPCR decreased from 97.98% (singleplex) to 77.85% (duplex), and R^2^ decreased from 0.97 (singleplex) to 0.94 (duplex). The Ct value was 21 when a sample contained 10^5^ DNA copies. The *P. acanthamoeba* qPCR increased slightly in efficiency from 74.51% (singleplex) to 78.11% (duplex), and R^2^ decreased from 0.97 (singleplex) to 0.93 (duplex). The Ct value was 19 when a sample contained 10^5^ DNA copies.

#### 3.1.2. Limit of Detection

Synthetic controls were added to extracted DNA of placenta, stomach content and lung samples. Spiked samples were analysed to determine the limit of detection of the qPCR assays. The limit of detection was defined as the lowest amount of target DNA that could be detected in 95% of replicates [42,43]. Pan-Chlamydiales qPCR limits of detection were <62.5 DNA copies in placenta, stomach content and <125 DNA copies in lung DNA. Limits of detection of the *Chlamydia abortus*/*Chlamydia pecorum* qPCR assay were five DNA copies in placenta and stomach content, and two DNA copies in lung DNA. Limits of detection for the *P. acanthamoeba*/*W. chondrophila* qPCR assay were <1 DNA copy (*P. acanthamoeba*) and 1 DNA copy (*W. chondrophila*) in placenta, <1 DNA copy in stomach content and <1 DNA copy in lung (Appendix A).

#### 3.1.3. Analytical Specificity

None of the three qPCR assays detected any of the non-target organisms. 

### 3.2. Clinical Samples

Fifty clinical samples with necrotic placentitis and/or pneumonia lesions from 25 abortion cases (17 placentas, 19 stomach contents and 14 lungs) were analysed. Animal species involved were bovine (*n* = 17), ovine (*n* = 6) and caprine (*n* = 2) (Table 3). In 16 cases, Chlamydiales were detected in at least one sample. In four of these cases, Chlamydiales were detected in two samples each. In two cases, *Chlamydia* had been detected previously by immunohistochemistry [36] and, in this study, Chlamydiales were again detected using qPCR. In eight cases, no Chlamydiales were detected in this study or the previous study by Jonker et al. [41]. A cut-off point (10^6^ DNA copies) experimentally determined by Gutierrez et al. [37] was applied to *C. abortus* results to determine diagnostic significance of results.

Chlamydiales were detected in ten bovine, four ovine and two caprine cases and most often in placenta (*n* = 12). The number of DNA copies was highest in placenta. Where Chlamydiales DNA was detected in more than one sample in a case, DNA copies were lower in stomach content and lung than in placenta. *C. abortus* was detected in three bovine cases only. *C. pecorum* was only detected in ovine and caprine cases. *P. acanthamoeba* was detected in three bovine cases. *W. chondrophila* was detected in bovine, ovine and caprine cases. *Chlamydia pecorum* (*n* = 2) and *W*. *chondrophila* (*n* = 1) were detected in cases where pneumonia lesions had been reported previously.

One case that was previously positive for *C. abortus* by PCR was not positive in this study. In one bovine and one caprine case, Chlamydiales could not be identified further by the assays in this study.

Seven cases had pathological lesions (necrotic placentitis (*n* = 5); pneumonia (*n* = 2)) that pointed to bacterial infection, but previously no agent could be detected [36]. In this study, Chlamydiales were detected in samples from three of these cases. *Chlamydia pecorum* was detected in this study in one case where *Chlamydia* species were detected previously by IHC. In four cases, no agent of abortion could be detected in this study and the previous study by Jonker et al. [36]. 

β-actin was detected in all samples, indicating efficient extraction and absence of inhibition factors that could lead to false negative results. 

#### 3.2.1. Diagnostic Sensitivity

Diagnostic sensitivity was calculated using the data in Table 3. The results were as follows: Pan-Chlamydiales assay: 0.38 (38%); *C. abortus*/*C. pecorum* assay: 0.2 (20%); *P. acanthamoeba*/*W*. *chondrophila* assay: 0.22 (22%). 

#### 3.2.2. Diagnostic Specificity

Forty-nine samples of placenta, stomach content and lung from bovine (*n* = 19), ovine (*n* = 9) and caprine (*n* = 10) cases where no placentitis or pneumonia was observed, were analysed using the qPCR assays in this study (Appendix A). In 45 samples either no target DNA was detected or Ct values higher than 32 were returned. Chlamydiales were detected at Ct values lower than 32 in five samples. Results of diagnostic specificity calculations were as follows: Pan-Chlamydiales assay: 0.67 (67.3%); *C. abortus*/*C. pecorum* assay: *C. abortus*: 0.94 (94%) and *C. pecorum*: 1 (100%); *P. acanthamoeba*/*W. chondrophila* assay: *P. acanthamoeba*: 0.18 (18.4%) and *W. chondrophila*: 1 (100%). 

## 4. Discussion

The aim of this study was to optimise and apply qPCR assays for detection of members of the order Chlamydiales and differentiation of *C. abortus*, *C. pecorum*, *P. acanthamoeba* and *W. chondrophila* in products of abortion, with necrotic placentitis and/or pneumonia lesions, from domestic ruminants in order to improve detection of intracellular bacteria that cannot be cultured on acellular media. Primers and probes for detection of DNA of the order Chlamydiales and of the species *C. abortus*, *C. pecorum*, *P. acanthamoeba* and *W. chondrophila* were selected from the literature. One singleplex and two duplex qPCR assays were optimised. These assays were applied to 50 samples from 25 clinical abortion cases, where necrotic placentitis and/or pneumonia were described histopathologically. An additional 49 samples where no pathological lesions were reported or where lesions other than placentitis and pneumonia were reported were also analysed using the three qPCR assays.

Slight modification of the Pan-Chlamydiales qPCR using a MGB probe instead of locked nucleic acids resulted in a correlation coefficient (R^2^) of 0.959 and an efficiency of 104.4%. Although efficiency lies between 90 and 105% [43], R^2^ was only 0.959, indicating that the serial dilution of the control was not accurately done [43]. The efficiency of this screening assay was much lower than the *C. abortus*/*C. pecorum* assay. This could be due to the use of a fragment of 16S RNA for the design of the primers and probe. Consequently, the Pan-Chlamydiales assay could miss positive cases that would be detected by the *C. abortus*/*C. pecorum* assay. Improvement of this assay should be attempted by amplifying a fragment from a different gene such as *ompA*. The *C. abortus*/*C. pecorum* assay had a high efficiency percentage at 99.5% (*C. abortus*) and 99% (*C. pecorum*) and R^2^ of 0.995 and 0.99 for *C. abortus* and *C. pecorum,* respectively. These results were slightly better than results when the assays were run in singleplex fashion. Efficiency of the *W*. *chondrophila* qPCR, which was 97.98% as a singleplex assay, decreased in the duplex assay, while efficiency of the *P. acanthamoeba* qPCR improved slightly (*P. acanthamoeba*: 78.11% and *W. chondrophila*: 77.86%). The use of an MGB probe improved the efficiency of this assay slightly; however, detection of targets was still impaired [42]. Improvement of the efficiency of the *P. acanthamoeba*/*W. chondrophila* qPCR assays must be attempted in future studies. Similar to the Pan-Chlamydiales assay, amplifying a fragment from a different gene such as *ompA* must be investigated to improve the assay. Improvement is essential if this assay is to be used for quantification studies in future since PCR efficiency plays an important role in quantification of target DNA [43]. 

Direct diagnosis of Chlamydiales consists of demonstration of microorganisms in tissues by methods such as the Modified Ziehl Neelsen staining method and IHC [26]. Chlamydiales were detected in 16 of the 25 clinical cases with histopathological lesions of necrotic placentitis or pneumonia. These results provide a preliminary indication of diagnostic sensitivity of the assays in this study, because Chlamydiales were detected in DNA from samples where histopathological lesions that are associated with infection by these bacteria were reported [4]. The calculated diagnostic sensitivity was quite low for all three assays, indicating either that the inclusivity of the PCR was low or that the limit of detection of the reference method was higher than the tested qPCR. Unfortunately, results of previous IHC or PCR analyses for Chlamydiales were only available in three cases. In two of these cases, Chlamydiales were also detected in this study. Further analysis of diagnostic sensitivity of the qPCR assays in this study by comparison with culture, IHC and other PCR methods is necessary. The most sensitive combination of methods in a diagnostic setting needs to be determined.

Analysis of samples from cases without lesions resulted in several Ct values higher than 32. Since there were no histopathological lesions, these results are more likely to be an indication of contamination. Diagnostic specificity results of the *C*. *abortus*/*C. pecorum* assay were highest, indicating good ability to identify samples found negative by the reference method. The results for the Pan-Chlamydiales and *P*. *acanthamoeba*/*W. chondrophila* assays were lower, indicating poorer exclusivity of the assays [43]. Another explanation is that the sensitivity of the reference method (histopathology) was low and the assays could detect more positive samples [43]. In addition to the reproductive tract, Chlamydiales can also colonise the intestinal tract [23]. Elementary bodies, the environmentally stable, infectious phase of the Chlamydiales developmental cycle [10], are shed into the environment together with products of abortion or faeces. Environmental contamination of placentae and foetuses can also lead to positive PCR results [14,23]. Therefore, the interpretation of PCR results by the laboratory diagnostician is of major importance. qPCR includes the option to quantify the amount of an agent present in a sample [14], which can be helpful in the discrimination between aetiologically significant pathogens and contaminants to prevent overestimation of the significance of a result.

A singleplex qPCR only detects one pathogen at a time. Several singleplex qPCR assays may be performed to detect different pathogens, but this can become quite costly and time consuming [44]. The minimum number of tests is often selected for economic reasons. For this reason, results of abortion investigations are often inconclusive [31]. A multiplex qPCR, on the other hand, can screen samples for several different pathogens at the same time, allowing the development of a more time and cost-effective analysis [44]. In this study, detection of potential agents of abortion improved from 19 to 21 out of 25 cases when multiplex qPCR was used and sample bias was removed by analysing all samples with necrotic placentitis and/or pneumonia lesions. 

The assays in this study enabled the detection of *C. pecorum* in two ovine cases. In both cases pneumonia was reported, which is not the case in reports by other authors [13,15]. The role of *C. pecorum* in ovine abortion was investigated by an Australian study [15] and it was found to be significant in some herds. In South Africa, the significance of this *Chlamydia* species is currently unknown and its investigation is warranted.

*Waddlia chondrophila* was detected in five bovine, three ovine and one caprine case, and *P. acanthamoeba* in two bovine cases in this study. *P. acanthamoeba* and *W*. *chondrophila* were detected before in samples from bovine, ovine and caprine abortion cases in countries in the Northern Hemisphere [16,24,33,45]. To the best of the authors’ knowledge, this is the first report of detection of *P. acanthamoeba* and *W. chondrophila* in South Africa. The role of these members of Chlamydiales as agents of domestic ruminant abortion in South Africa is currently unknown.

In this study, like many others, several co-infections were detected. In an effort to differentiate between infection and colonisation or contamination, a diagnostic cut-off point (10^6^ DNA copies) experimentally determined for *C. abortus* by Gutierrez et al. [37] was applied. In this study, this cut-off point excluded all cases where *C. abortus* was detected except for one bovine case. However, there may be fewer than 10^6^ DNA copies of a causative agent in a sample due to sampling or processing. So, although a diagnostic cut-off point can be a useful tool it can also not be used in isolation to determine the significance of a qPCR result. A combination of analyses such as histopathology, IHC and qPCR as well as evaluation of all results by the laboratory diagnostician is once again essential to attach appropriate significance to detection of Chlamydiales. Since *C. abortus* is not commonly associated with bovine abortion it may not be selected from a list of singleplex PCR options. In this study, the *C. abortus* was detected, because the analysis was included in a multiplex assay targeting bacteria that cause necrotic placentitis or pneumonia and test selection was not possible. This illustrates the advantage of a multiplex assay that can detect several agents associated with a particular pathological lesion in one reaction.

Chlamydiales were detected in samples from two ovine cases, *C. pecorum* in one case and *W. chondrophila* in the other, where only pathological lesions pointing to an infectious cause were reported previously by Jonker et al. [41]. Lack of any other possible agents of abortion increases their significance in these cases. However, the efficiency of the *P. acanthamoeba*/*W. chondrophila* qPCR used in this study was considered too low to aid in the diagnosis, indicating the need for the design of new assays.

In one case, *C. abortus* was detected previously, but not in this study. The reason may be that a different sample from the same case was used and that there was not enough DNA in the sample. Chlamydiales detected in three bovine and one caprine case could not be identified further by the assays in this study. Internationally, more than one author has reported Chlamydiales they could not identify [31,46]. This serves as an indication that other Chlamydiales may be agents of abortion. Further investigation by PCR and sequencing is necessary to identify these agents and determine their role in abortions. The findings in this study indicate that further investigation of the role of Chlamydiales in ruminant abortion in South Africa is warranted.

## 5. Conclusions

Application of the three qPCR assays in this study improved detection of members of the order Chlamydiales in samples from cases with necrotic placentitis and/or pneumonia. In addition, differentiation of the Chlamydiales to species level was improved. Application of a diagnostic cut-off point was useful to identify clinically significant Ct values for *C. abortus*. Detection of *C. pecorum, P. acanthamoebae* and *W. chondrophila* in ruminant abortion cases in South Africa indicate that these bacteria potentially play a role in abortions. 

It is recommended that, in future, these qPCR assays form part of a comprehensive diagnostic approach to ruminant abortions. These results serve as a reminder that ruminant abortion material should be handled with appropriate safety precautions since *C. abortus*, *P. acanthamoebae* and *W. chondrophila* are potentially zoonotic agents.

## Figures and Tables

**Figure 1 pathogens-12-00290-f001:**
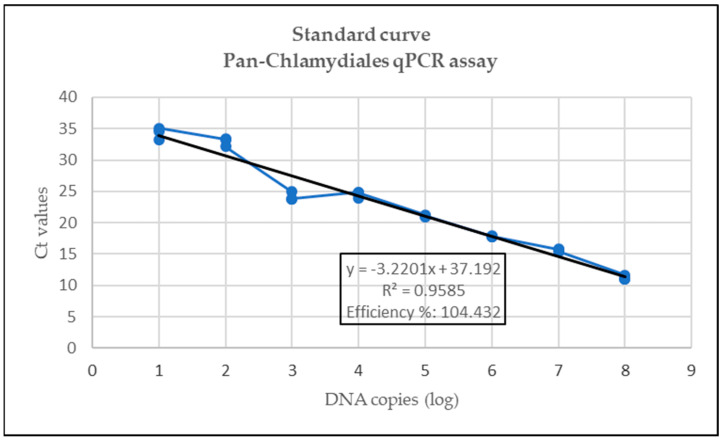
Standard curve for the Pan-Chlamydiales assay where Ct values (*y*-axis) are plotted against the log of the concentration of DNA copies (*x*-axis).

**Figure 2 pathogens-12-00290-f002:**
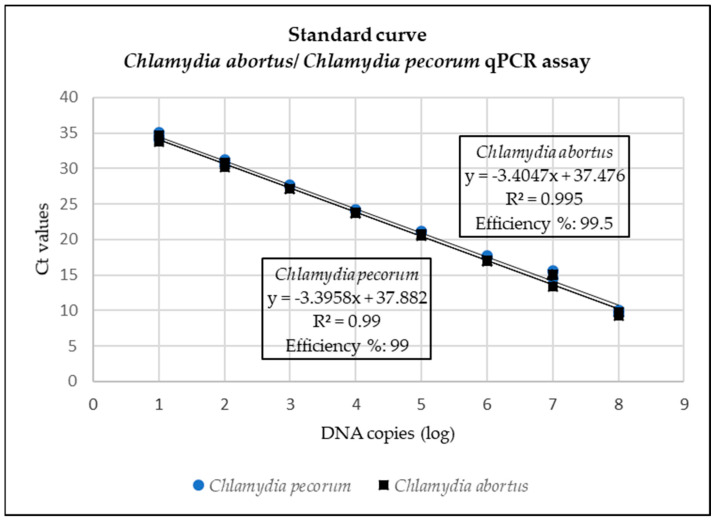
Standard curve for the *Chlamydia abortus*/*Chlamydia pecorum* assay where Ct values (*y*-axis) are plotted against the log of the concentration of DNA copies (*x*-axis).

**Figure 3 pathogens-12-00290-f003:**
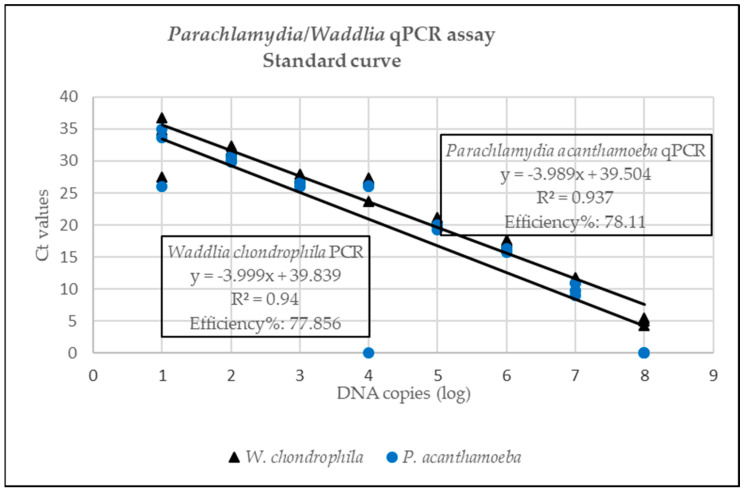
Standard curve for the *Parachlamydia acanthamoeba*/*Waddlia acanthamoeba* duplex qPCR where Ct values (*y*-axis) are plotted against the log of the concentration of DNA copies (*x*-axis).

**Table 1 pathogens-12-00290-t001:** Bacteria and amoeba species used as negative controls to test specificity.

Bacteria	Source or Strain
*Acholeplasma laidlawlii*	NCTC 10116
*Brucella abortus*	Clinical sample
*Chlamydia abortus*	ATCC VR-165
*Escherichia coli*	ATCC 25922
*Ochrobactrum anthropi*	ATCC 49687
*Parachlamydia acanthamoeba*	ATCC VR-1476
*Pasteurella multocida*	ATCC 12945
*Salmonella Typhimurium*	ATCC 13311
*Staphylococcus aureus*	ATCC 25923
*Streptococcus agalactiae*	ATCC 27956
*Trueperella pyogenes*	Clinical sample
*Waddlia chondrophila*	ATCC VR-1470
**Amoeba**	
*Acanthamoeba castellani*	ATCC 50739

**Table 3 pathogens-12-00290-t003:** Chlamydiales DNA detected in bovine, ovine and caprine cases in this study together with the samples where they were detected, necropsy lesions, bacteria detected previously and cycle thresholds. (ND = not detected, Ct = cycle threshold, *PanChl* = Pan-Chlamydiales, *Chlab* = *Chlamydia abortus*, *Chlpec* = *Chlamydia pecorum*, *Pacanth* = *Parachlamydia acanthamoeba*, *Wchon* = *Waddlia chondrophila*).

Case No.	Pathology	Bacteria/Fungi (Previous Study [41])	Samples	DNA Detected (Mean Ct Values)
**Bovine**				
1	Necrotic placentitis Purulent bronchopneumonia	*Bacillus licheniformis* *Rhizopus* sp.	Stomach content	ND
			Lung	ND
2	Necrotic placentitis and vasculitis	*Mycoplasma* sp. *Aspergillus fumigatus*	Placenta	*PanChl* (36.05)
			Stomach content	*PanChl* (33.38) *Chlab* (32.33) *Chlpec* (36.90) *Wchon* (31.85)
			Lung	ND
3	Necrotic placentitis	ND	Stomach content	ND
4	Necrotic placentitis	*Salmonella* sp.	Placenta	*PanChl* (32.90) *Wchon* (26.02)
5	Necrotic placentitis with vasculitis	*Coxiella burnetii*	Stomach content	*PanChl* (32.25) *Chlab* (32.18) *Chlpec* (35.86)
6	Necrotic placentitis	*Streptococcus dysgalactiae* *Acholeplasma* sp.	Placenta	*PanChl* (28.63)
			Stomach content	*Wchon* (34.36)
			Lung	ND
7	Purulent placentitis	ND	Placenta	ND
8	Necrotic placentitis	*Lichtheimia corymbifera*	Placenta	*PanChl* (31.21) *Wchon* (23.97)
			Stomach content	ND
			Lung	ND
9	Meningitis bronchointerstitial pneumonia	*Brucella abortus biovar 1*	Stomach content	ND
10	Meningoencephalitis Necrotic placentitis Hepatic necrosis	ND	Stomach content	ND
11	Necrotic placentitis with vasculitis	ND	Placenta	ND
		ND	Lung	ND
12	Necrotic placentitis	*Campylobacter fetus* *Mannheimia varigena* *Streptococcus pluranimalium* *Mycoplasma* sp.	Placenta	*PanChl* (30.02) *Pacanth* (36.88) *Wchon* (32.47)
			Stomach content	ND
			Lung	ND
13	Necrotic placentitis Purulent pneumonia	*Chlamydia* sp. *Salmonella* sp.	Placenta	*PanChl* (31.63)
			Stomach content	*PanChl* (33.94)
			Lung	ND
14	Necrotic placentitis	ND	Placenta	*PanChl* (5.03) *Chlab* (5.65)
			Stomach content	ND
15	Viral placentitis	*Pasteurella multocida*	Placenta	*Chlab* (30.79)
16	Interstitial pneumonia	*Brucella abortus* biovar 1	Stomach content	ND
17	Necropurulent placentitis	*Trueperella pyogenes*	Placenta	*PanChl* (34.68) *Pacanth* (33.48)
18	Necrotic placentitis Pneumonia	*Chlamydia* sp.	Placenta	*PanChl* (23.13) *Chlpec* (18.96)
			Stomach content	*PanChl* (31.69) *Chlpec* (29.83)
19	Bacterial & fungal placentitis Pleuropneumonia	*Chlamydia abortus*	Stomach content	ND
			Lung	ND
20	Bronchopneumonia	ND	Stomach content	*PanChl* (27.42) *Chlpec* (22.19) *Wchon* (35.17)
			Lung	*PanChl* (25.74) *Chlpec* (23.42) *Wchon* (35.39)
21	Bacterial placentitis	*Arcobacter* sp.	Placenta	ND
			Lung	ND
22	Necrotic placentitis	*Salmonella* Budapest *Mycoplasma* species	Placenta	*PanChl* (35.03) *Wchon* (33.35)
			Stomach content	ND
23	Fibrinopurulent bacterial pneumonia	ND	Placenta	*PanChl* (29.49) *Chlpec* (34.48) *Wchon* (32.52)
			Lung	ND
24	Necrotic placentitis	*Coxiella burnetiid* *Salmonella typhimurium*	Placenta	ND
			Stomach content Foetus A	ND
			Lung Foetus A	*PanChl* (32.41) *Chlpec* (35.84) *Wchon* (31.57)
			Stomach content Foetus B	ND
			Lung Foetus B	ND
25	Necrotizing placentitis Subcutaneous oedema Hydrocephalus	*E. coli* *Lichtheimia corymbifera*	Placenta	*PanChl* (10.27)
			Stomach content	ND
			Lung	ND

## Data Availability

Data are contained within the article.

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
