# Peer review of "Optimization and Application of Real-Time qPCR Assays in Detection and Identification of Chlamydiales in Products of Domestic Ruminant Abortion"

_pathogens, 2023, doi:10.3390/pathogens12020290_

Round 1

Reviewer 1 Report

The authors present a diagnostic approach to detect Chlamydiales in ruminant abortion products with the final aim to decide on the etiological role of the detected pathogens within the abortive process. The manuscript is very well organized and written in a clear language. Further, the first report on the emergence of Parachlamydia acanthamoeba and Waddlia chondrophila in South African ruminants is an interesting detail.

However, the study approach is not particularly novel or original since established qPCR protocols were used and there is a general lack of rigor in the methodology and data analysis. Indeed, for the diagnostic testing, low sample numbers are used, only partial validation is presented, no definition of gold standards, no calculation of sensitivities or specificities, no analysis of statistical significance, and finding/definition of LoD is completely unclear (see also comments below).

I see a gap between the claims in the title and the data presented, which does not fully support them. The improvements compared to a previous study of the authors with the same sample set are small, especially the performance of the qPCR assays was not always convincing with very high LoD for Chlamydiales-qPCR and low efficiency for Waddlia/Parachlamydia-qPCR. Unfortunately, the authors did not discuss or try to optimize that.

Please see my specific comment (major and minor) below:

Introduction:

L 35-37:Chlamydia abortus (C. abortus), Waddlia chondrophila (W. chondrophila) and Parachlamydia acanthamoebae (P. acanthamoebae) are associated with human miscarriages [5, 6]. In addition, P. acanthamoebae is an emerging agent of pneumonia in humans [7].”

This statement is too strong. The zoonotic implications of this pathogens are still rather vague.

L23: give more references for molecular methods

L57 and throughout the manuscript: change “chondrophilia” in “”chondrophila”

L84: the study from Ref27 was from Switzerland, not from Germany

Materials and Methods:

Table 1: bad table layout

L154-159: The quantification of bacterial culture prior to DNA preparation and PCR testing is not clearly explained. Acholeplasma, for instance cannot be quantified by the McFarland standard.

Clinical samples

It is not clear to the reader why only 32 cases of abortion without placentitis/pneumonia and 25 with placentitis/ pneumonia were selected from 135 cases. What about the rest of 78 cases? Why were they not elegible?

L172: change “was” in “were” twice

L174-176: give amounts of sample tissue and bacterial culture used for DNA extraction

L182: change “Thermofischer” in “Thermo Fisher”

Table 2: bad table layout

Use of two FAM probes in Chlamydia-specific duplex PCR does not work, different fluorophores have to be used.

L196: change “20µl” in “20 µl”

L202: change “95°C for 15 sec for 40 cycles (denaturation) and 60°C for 1 min (annealing/elongation).” in “and 40 cycles of 95°C for 15 sec and 60°C for 1 min (annealing/elongation).”

Results:

3.1.1. Analytical sensitivity

It would be good to compare efficiency of simplex and duplex runs

Pan Chlamydiales qPCR and Parachlamydia/Waddlia duples qPCR are obviously not linear over the concentration range 10^1-10^8 GE/µl. This should be mentioned and considered for efficiency and R^2 calculations.

3.1.2. Limit of detection

It is not shown how the LoD was exactly determined. Which Ct values were obtained in the dilution series and how, by which criteria the cut off/LoD was defined?

The statement in L265-267 does not give a comprehensible explanation.

In my experience it is difficult to spike samples with standard DNA prior to sample DNA extraction because the DNA standards are often destroyed by the DNA preparation procedure. Didn’t you meet these problems?

Table 3: Does not give additional information to the text

Table 4: very bad table layout, has to be improved before online publication.

It is not clear how a sample with Ct between 32 and 33 is categorized (suspect or inconclusive?). Please correct this in the table.

The Ct value obtained for each sample should be included in the table. Results should also be shown for negative samples (without lesions).

Discussion:

To evaluate optimization success for qPCR assay, they have to be run in one lab under the same conditions, otherwise comparison of efficiencies and R^2 values makes no sense.

It should be discussed why the Chlamydiales qPCR is so much less sensitive compared to the other qPCR assays and what could be improved.

The efficiencies of the Parachlamydia and Waddlia qPCRs are too low. I would not consider them for diagnostic purposes but try to design new assays. For quantification and diagnostic purposes the PCR should exhibit efficiencies between 90 and 110 %.

L361: environmental contamination with Chlamydiales does not lead to false-positive PCR results, but to true-positive PCR results, because PCR as a method cannot discriminate between etiological significance and contamination. Rather, the interpretation of PCR results by the investigator/veterinarian is of high importance. Please change wording here.

L386ff.: The diagnostic cut off by Gutierrez et al was determined only for the C. abortus PCR and does not automatically apply for the other protocols used in this study.

L417: add “in” before “South Africa”

Conclusion:

L426: add ”,” after “in future”

Reviewer 2 Report

The study describes the optimisation of a molecular technique for the detection of Chlamydiales members. The study is methodologically correct and well written. There a few points that need correction however before being considered for publication.

The title does not fully reflect the findings and the scope of the study. Apart from the optimisation of the available molecular techniques and development of singleplex and two duplex assays, P. acanthamoeba and W. chondrophilia were identified, whereas the authors propose a role of these pathogens in ruminant abortions. I therefore suggest to the authors to modify the title accordingly. Also the abstract is a little confusing. It should be separated according to the clinical findings and the results of the developed technique. 

In line 54 the authors should change paragraph. As it is, it misses connectivity.

Lines 82-84 the statement is not completely correct. qPCR indeed achieves quantification of the target, however it does not provide increased sensitivity. Real time PCR does so. qPCR is only a subcategory of real time PCR. Please correct

Lines 132-143: Please provide the sequences of the synthetic controls as supplementary files

Line 199: “were increased” to what concentration? Please specify

How the discrepancies referred in lines 308-316 could be explained? Please propose detailed explanations 

Round 2

Reviewer 1 Report

The authors have addressed most of the critical points raised and improved the overall impression of the paper.

However, I strongly recommend a few changes:

The desription and presentation of LOD determination is still not satisfying. Figures 4-12 are not informative and unnecessary. The information is much better presented in the supplement tables S2-S12. Please, give rounded Ct values in the format X.YZ or X.Y there.

From the results in the supplementary table, I cannot deduce the LOD numbers given by the authors. According to LOD definition given by the authors ("The limit of detection was defined as the lowest amount of target DNA that could be detected in 95% of replicates [42, 43].), LOD for pan-Chlamydiales PCR in placenta should be <62.5 DNA copies and not 500 (although it is strange that Ct for 500 copies is 34 and for 62.6 copies 30 ???) or 2 copies for C. abortus PCR in lung and so on. Please, revise critically.

Further, a diagnostic cut off of 10^6 copies with an equivalent of Ct16.97 in your PCR assay seems very high (or low concerning the Ct) and not realistic. In my experience, Ct 20-25 can be good indications for a C. abortus involvement in abortion, al least in sheep.

Ct values of 3,11 (table 4 sample 14, L476) are very unusual and for sure out of the linear amplification range. Are you sure they are valid?
